# Discovery of Latent 3D Keypoints via End-to-end Geometric Reasoning

**Supasorn Suwajanakorn**$^{\triangleright *}$    **Noah Snavely**$^{\diamond}$    **Jonathan Tompson**$^{\diamond}$    **Mohammad Norouzi**$^{\diamond}$

supasorn@vistec.ac.th, {snavely, tompson, mnorouzi}@google.com

$^{\triangleright}$Vidyasirimedhi Institute of Science and Technology    $^{\diamond}$Google AI

## Abstract

This paper presents *KeypointNet*, an end-to-end geometric reasoning framework to *learn* an optimal set of *category-specific* 3D keypoints, along with their detectors. Given a single image, KeypointNet extracts 3D keypoints that are optimized for a downstream task. We demonstrate this framework on 3D pose estimation by proposing a differentiable objective that seeks the optimal set of keypoints for recovering the relative pose between two views of an object. Our model discovers geometrically and semantically consistent keypoints across viewing angles and instances of an object category. Importantly, we find that our end-to-end framework using no ground-truth keypoint annotations outperforms a fully supervised baseline using the same neural network architecture on the task of pose estimation. The discovered 3D keypoints on the car, chair, and plane categories of ShapeNet [6] are visualized at *keypointnet.github.io*.

## 1  Introduction

Convolutional neural networks have shown that jointly optimizing feature extraction and classification pipelines can significantly improve object recognition [26, 25]. That being said, current approaches to geometric vision problems, such as 3D reconstruction [47] and shape alignment [29], comprise a separate *keypoint* detection module, followed by geometric reasoning as a post-process. In this paper, we explore whether one can benefit from an *end-to-end* geometric reasoning framework, in which keypoints are jointly optimized as a set of *latent variables* for a downstream task.

Consider the problem of determining the 3D pose of a car in an image. A standard solution first detects a sparse set of category-specific keypoints, and then uses such points within a geometric reasoning framework (*e.g.,* a PnP algorithm [28]) to recover the 3D pose or camera angle. Towards this end, one can develop a set of keypoint detectors by leveraging strong supervision in the form of manual keypoint annotations in different images of an object category, or by using expensive and error prone offline model-based fitting methods. Researchers have compiled large datasets of annotated keypoints for faces [44], hands [51], and human bodies [3, 30]. However, selection and consistent annotation of keypoints in images of an object category is expensive and ill-defined. To devise a reasonable set of points, one should take into account the downstream task of interest. Directly optimizing keypoints for a downstream geometric task should naturally encourage desirable keypoint properties such as distinctiveness, ease of detection, diversity, *etc.*

This paper presents *KeypointNet*, an end-to-end geometric reasoning framework to learn an *optimal* set of category-specific 3D keypoints, along with their detectors, for a specific downstream task. Our novelty stands in contrast to prior work that learns latent keypoints through an arbitrary proxy self-supervision objective, such as reconstruction [63, 17]. Our framework is applicable to any downstream task represented by an objective function that is differentiable with respect to keypoint positions. We formulate 3D pose estimation as one such task, and our key technical contributions include (1) a novel differentiable pose estimation objective and (2) a multi-view consistency loss

---

$^{*}$Work done while S. Suwajanakorn was a member of the Google AI Residency program (g.co/airesidency).

function. The pose objective seeks optimal keypoints for recovering the relative pose between two views of an object. The multi-view consistency loss encourages consistent keypoint detections across 3D transformations of an object. Notably, we propose to detect *3D* keypoints (2D points with depth) from individual 2D images and formulate pose and consistency losses for such 3D keypoint detections.

We show that KeypointNet discovers geometrically and semantically consistent keypoints across viewing angles as well as across object instances of a given class. Some of the discovered keypoints correspond to interesting and semantically meaningful parts, such as the wheels of a car, and we show how these 3D keypoints can infer their depths without access to object geometry. We conduct three sets of experiments on different object categories from the ShapeNet dataset [6]. We evaluate our technique against a strongly supervised baseline based on manually annotated keypoints on the task of relative 3D pose estimation. Surprisingly, we find that our end-to-end framework achieves significantly better results, despite the lack of keypoint annotations.

## 2 Related Work

Both 2D and 3D keypoint detection are long-standing problems in computer vision, where keypoint inference is traditionally used as an early stage in object localization pipelines [27]. As an example, a successful early application of modern convolutional neural networks (CNNs) was on detecting 2D human joint positions from monocular RGB images. Due to its compelling utility for HCI, motion capture, and security applications, a large body of work has since developed in this joint detection domain [53, 52, 39, 37, 62, 38, 20, 14].

More related to our work, a number of recent CNN-based techniques have been developed for 3D human keypoint detection from monocular RGB images, which use various architectures, supervised objectives, and 3D structural priors to directly infer a predefined set of 3D joint locations [36, 34, 8, 35, 12]. Other techniques use inferred 2D keypoint detectors and learned 3D priors to perform "2D-to-3D-lifting" [41, 7, 66, 33] or find data-to-model correspondences from depth images [40]. Honari *et al.* [18] improve landmark localization by incorporating semi-supervised tasks such as attribute prediction and equivariant landmark prediction. In contrast, our set of keypoints is not defined *a priori* and is instead a latent set that is optimized end-to-end to improve inference for a geometric estimation problem. A body of work also exists for more generalized, albeit supervised, keypoint detection, *e.g.,* [15, 61].

Enforcing latent structure in CNN feature representations has been explored for a number of domains. For instance, the *capsule* framework [17] and its variants [43, 16] encode activation properties in the magnitude and direction of hidden-state vectors and then combine them to build higher-level features. The output of our KeypointNet can be seen as a similar form of latent 3D feature, which is encouraged to represent a set of 3D keypoint positions due to the carefully constructed consistency and relative pose objective functions.

Recent work has demonstrated 2D correspondence matching across intra-class instances with large shape and appearance variation. For instance, Choy *et al.* [9] use a novel contrastive loss based on appearance to encode geometry and semantic similarity. Han *et al.* [13] propose a novel SCNet architecture for learning a geometrically plausible model for 2D semantic correspondence. Wang *et al.* [60] rely on deep features and perform a multi-image matching across an image collection by solving a feature selection and labeling problem. Thewlis *et al.* [49] use ground-truth transforms (optical flow between image pairs) and point-wise matching to learn a dense object-centric coordinate frame with viewpoint and image deformation invariance. Similarly, Agrawal *et al.* [2] use egomotion prediction between image pairs to learn semi-supervised feature representations, and show that these features are competitive with supervised features for a variety of tasks.

Other work has sought to learn latent 2D or 3D features with varying amounts of supervision. Arie-Nachimson & Basri [4] build 3D models of rigid objects and exploit these models to estimate 3D pose from a 2D image as well as a collection of 3D latent features and visibility properties. Inspired by cycle consistency for learning correspondence [19, 65], Zhou *et al.* [64] train a CNN to predict correspondence between different objects of the same semantic class by utilizing CAD models. Independent from our work, Zhang *et al.* [63] discover sparse 2D landmarks of images of a known object class as explicit structure representation through a reconstruction objective. Similarly, Jakab and Gupta *et al.* [23] use conditional image generation and reconstruction objective to learn 2D keypoints that capture geometric changes in training image pairs. Rhodin *et al.* [42] uses a

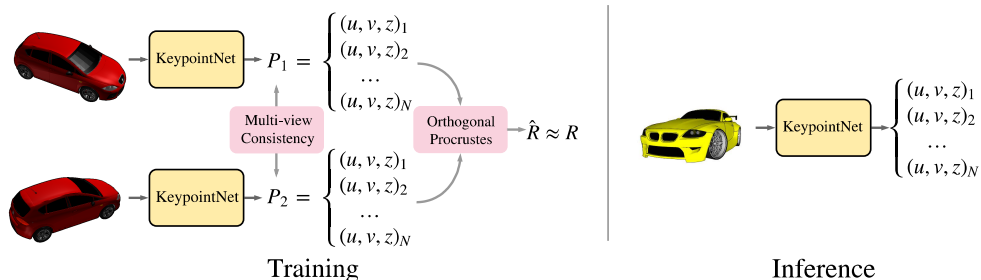

Training | Inference

Figure 1: During training, two views of the same object are given as input to the KeypointNet. The known rigid transformation $(R, t)$ between the two views is provided as a supervisory signal. We optimize an ordered list of 3D keypoints that are consistent in both views and enable recovery of the transformation. During inference, KeypointNet extracts 3D keypoints from an individual input image.

multi-view consistency loss, similar to ours, to infer 3D latent variables specifically for human pose estimation task. In contrast to [64, 63, 23, 42], our latent keypoints are optimized for a downstream task, which encourages more directed keypoint selection. By representing keypoints in true physical 3D structures, our method can even find occluded correspondences between images with large pose differences, *e.g.,* large out-of-plane rotations.

Approaches for finding 3D correspondence have been investigated. Salti *et al.* [45] cast 3D keypoint detection as a binary classification between points whose ground-truth similarity label is determined by a predefined 3D descriptor. Zhou *et al.* [67] use view-consistency as a supervisory signal to predict 3D keypoints, although only on depth maps. Similarly, Su *et al.* [48] leverage synthetically rendered models to estimate object viewpoint by matching them to real-world image via CNN viewpoint embedding. Besides keypoints, self-supervision based on geometric and motion reasoning has been used to predict other forms of output, such as 3D shape represented as blendshape coefficients for human motion capture [57].

## 3   End-to-end Optimization of 3D Keypoints

Given a single image of a known object category, our model predicts an ordered list of 3D keypoints, defined as pixel coordinates and associated depth values. Such keypoints are required to be geometrically and semantically consistent across different viewing angles and instances of an object category (*e.g.,* see Figure 4). Our KeypointNet has $N$ heads that extract $N$ keypoints, and the same head tends to extract 3D points with the same semantic interpretation. These keypoints will serve as a building block for feature representations based on a sparse set of points, useful for geometric reasoning and pose-aware or pose-invariant object recognition (*e.g.,* [43]).

In contrast to approaches that learn a supervised mapping from images to a list of annotated keypoint positions, we do not define the keypoint positions *a priori*. Instead, we jointly optimize keypoints with respect to a downstream task. We focus on the task of *relative pose estimation* at training time, where given two views of the same object with a known rigid transformation $T$, we aim to predict optimal lists of 3D keypoints, $P_1$ and $P_2$ in the two views that best match one view to the other (Figure 1). We formulate an objective function $O(P_1, P_2)$, based on which one can optimize a parametric mapping from an image to a list of keypoints. Our objective consists of two primary components:

- A *multi-view consistency* loss that measures the discrepancy between the two sets of points under the ground truth transformation.
- A *relative pose estimation* loss, which penalizes the angular difference between the ground truth rotation $R$ *vs.* the rotation $\hat{R}$ recovered from $P_1$ and $P_2$ using orthogonal procrustes.

We demonstrate that these two terms allow the model to discover important keypoints, some of which correspond to semantically meaningful locations that humans would naturally select for different object classes. Note that we do not directly optimize for keypoints that are semantically meaningful, as those may be sub-optimal for downstream tasks or simply hard to detect. In what follows, we first explain our objective function and then describe the neural architecture of KeypointNet.

**Notation.** Each training tuple comprises a pair of images $(I, I')$ of the same object from different viewpoints, along with their relative rigid transformation $T \in SE(3)$, which transforms the underlying 3D shape from $I$ to $I'$. $T$ has the following matrix form:

$$T = \begin{bmatrix} R^{3 \times 3} & t^{3 \times 1} \\ 0 & 1 \end{bmatrix}, \tag{1}$$

where $R$ and $t$ represent a 3D rotation and translation respectively. We learn a function $f_\theta(I)$, parametrized by $\theta$, that maps a 2D image $I$ to a list of 3D points $P = (p_1, \ldots, p_N)$ where $p_i \equiv (u_i, v_i, z_i)$, by optimizing an objective function of the form $O(f_\theta(I), f_\theta(I'))$.

## 3.1 Multi-view consistency

The goal of our multi-view consistency loss is to ensure that the keypoints track consistent parts across different views. Specifically, a 3D keypoint in one image should project onto the same pixel location as the corresponding keypoint in the second image. For this task, we assume a perspective camera model with a known global focal length $f$. Below, we use $[x, y, z]$ to denote 3D coordinates, and $[u, v]$ to denote pixel coordinates. The projection of a keypoint $[u, v, z]$ from image $I$ into image $I'$ (and vice versa) is given by the projection operators:

$$[\hat{u}, \hat{v}, \hat{z}, 1]^\top \quad \sim \quad \pi T \pi^{-1}([u, v, z, 1]^\top)$$
$$[\hat{u}', \hat{v}', \hat{z}', 1]^\top \quad \sim \quad \pi T^{-1} \pi^{-1}([u', v', z', 1]^\top)$$

where, for instance, $\hat{u}$ denotes the projection of $u$ to the second view, and $\hat{u}'$ denotes the projection of $u'$ to the first view. Here, $\pi : \mathbb{R}^4 \to \mathbb{R}^4$ represents the perspective projection operation that maps an input homogeneous 3D coordinate $[x, y, z, 1]^\top$ in camera coordinates to a pixel position plus depth:

$$\pi([x, y, z, 1]^\top) = \left[ \frac{fx}{z}, \frac{fy}{z}, z, 1 \right]^\top = [u, v, z, 1]^\top \tag{2}$$

We define a symmetric multi-view consistency loss as:

$$L_{\text{con}} = \frac{1}{2N} \sum_{i=1}^{N} \left\| [u_i, v_i, u_i', v_i']^\top - [\hat{u}_i', \hat{v}_i', \hat{u}_i, \hat{v}_i]^\top \right\|^2 \tag{3}$$

We measure error only in the observable image space $(u, v)$ as opposed to also using $z$, because depth is never directly observed, and usually has different units compared to $u$ and $v$. Note however that predicting $z$ is critical for us to be able to project points between the two views.

Enforcing multi-view consistency is sufficient to infer a consistent set of 2D keypoint positions (and depths) across different views. However, this consistency alone often leads to a degenerate solution where all keypoints collapse to a single location, which is not useful. One can encode an explicit notion of diversity to prevent collapsing, but there still exists infinitely many solutions that satisfy multi-view consistency. Rather, what we need is a notion of optimality for selecting keypoints which has to be defined with respect to some downstream task. For that purpose, we use pose estimation as a task which naturally encourages keypoint separation so as to yield well-posed estimation problems.

## 3.2 Relative pose estimation

One important application of keypoint detection is to recover the relative transformation between a given pair of images. Accordingly, we define a differentiable objective that measures the misfit between the estimated relative rotation $\hat{R}$ (computed via Procrustes' alignment of the two sets of keypoints) and the ground truth $R$. Given the translation equivariance property of our keypoint prediction network (Section 4) and the view consistency loss above, we omit the translation error in this objective. The pose estimation objective is defined as :

$$L_{\text{pose}} = 2 \arcsin \left( \frac{1}{2\sqrt{2}} \left\| \hat{R} - R \right\|_F \right) \tag{4}$$

which measures the angular distance between the optimal least-squares estimate $\hat{R}$ computed from the two sets of keypoints, and the ground truth relative rotation matrix $R$. Fortunately, we can formulate this objective in terms of fully differentiable operations.

To estimate $\hat{R}$, let $X$ and $X' \in \mathbb{R}^{3 \times N}$ denote two matrices comprising unprojected 3D keypoint coordinates for the two views. In other words, let $X \equiv [X_1, \ldots, X_N]$ and $X_i \equiv (\pi^{-1} p_i)[:3]$, where $[:3]$ returns the first 3 coordinates of its input. Similarly $X'$ denotes unprojected points in $P'$. Let $\tilde{X}$ and $\tilde{X}'$ denote the mean-subtracted version of $X$ and $X'$, respectively. The optimal least-squares rotation $\hat{R}$ between the two sets of keypoints is then given by:

$$\hat{R} = V \operatorname{diag}(1, 1, \ldots, \det(VU^\top)) U^\top, \tag{5}$$

where $U, \Sigma, V^\top = \operatorname{SVD}(\tilde{X}\tilde{X}'^\top)$. This estimation problem to recover $\hat{R}$ is known as the orthogonal Procrustes problem [46]. To ensure that $\tilde{X}\tilde{X}'^\top$ is invertible and to increase the robustness of the keypoints, we add Gaussian noise to the 3D coordinates of the keypoints ($X$ and $X'$) and instead seek the best rotation under some noisy predictions of keypoints. To minimize the angular distance (4), we backpropagate through the SVD operator using matrix calculus [22, 10].

Empirically, the pose estimation objective helps significantly in producing a reasonable and natural selection of latent keypoints, leading to the automatic discovery of interesting parts such as the wheels of a car, the cockpit and wings of a plane, or the legs and back of a chair. We believe this is because these parts are geometrically consistent within an object class (e.g., circular wheels appear in all cars), easy to track, and spatially varied, all of which improve the performance of the downstream task.

## 4 KeypointNet Architecture

One important property for the mapping from images to keypoints is translation *equivariance* at the pixel level. That is, if we shift the input image, *e.g.,* to the left by one pixel, the output locations of all keypoints should also be changed by one unit. Training a standard CNN without this property would require a larger training set that contains objects at every possible location, while still providing no equivariance guarantees at inference time.

We propose the following simple modifications to achieve equivariance. Instead of regressing directly to the coordinate values, we ask the network to output a probability distribution map $g_i(u, v)$ that represents how likely keypoint $i$ is to occur at pixel $(u, v)$, with $\sum_{u,v} g_i(u, v) = 1$. We use a spatial softmax layer to produce such a distribution over image pixels [11]. We then compute the expected values of these spatial distributions to recover a pixel coordinate:

$$[u_i, v_i]^\top = \sum_{u,v} [u \cdot g_i(u, v), v \cdot g_i(u, v)]^\top \tag{6}$$

For the $z$ coordinates, we also predict a depth value at every pixel, denoted $d_i(u, v)$, and compute

$$z_i = \sum_{u,v} d_i(u, v) g_i(u, v). \tag{7}$$

To produce a probability map with the same resolution and equivariance property, we use strided-one fully convolutional architectures [31], also used for semantic segmentation. To increase the receptive field of the network, we stack multiple layers of dilated convolutions, similar to [59].

Our emphasis on designing an equivariant network not only helps significantly reduce the number of training examples required to achieve good generalization, but also removes the computational burden of converting between two representations (spatial-encoded in image to value-encoded in coordinates) from the network, so that it can focus on other critical tasks such as inferring depth.

**Architecture details.** All kernels for all layers are $3 \times 3$, and we stack 13 layers of dilated convolutions with dilation rates of $1, 1, 2, 4, 8, 16, 1, 2, 4, 8, 16, 1, 1$, all with 64 output channels except the last layer which has $2N$ output channels, split between $g_i$ and $d_i$. We use leakyRelu and Batch Normalization [21] for all layers except the last layer. The output layers for $d_i$ have no activation function, and the channels are passed through a spatial softmax to produce $g_i$. Finally, $g_i$ and $d_i$ are then converted to actual coordinates $p_i$ using Equations (6) and (7).

**Breaking symmetry.** Many object classes are symmetric across at least one axis, *e.g.,* the left side of a sedan looks like the right side flipped. This presents a challenge to the network because different parts can appear visually identical, and can only be resolved by understanding global context. For example, distinguishing the left wheels from the right wheels requires knowing its orientation

(*i.e.,* whether it is facing left or right). Both supervised and unsupervised techniques benefit from some global conditioning to aid in breaking ties and to make the keypoint prediction more deterministic.

To help break symmetries, one can condition the keypoint prediction on some coarse quantization of the pose. Such a coarse-to-fine approach to keypoint detection is discussed in more depth in [56]. One simple such conditioning is a binary flag that indicates whether the dominant direction of an object is facing left or right. This dominant direction comes from the ShapeNet dataset we use (Section 6), where the 3D models are consistently oriented. To infer keypoints without this flag at inference time, we train a network with the same architecture, although half the size, to predict this binary flag.

In particular, we train this network to predict the projected pixel locations of two 3D points $[1, 0, 0]$ and $[-1, 0, 0]$, transformed into each view in a training pair. These points correspond to the front and back of a normalized object. This network has a single $L_2$ loss between the predicted and the ground-truth locations. The binary flag is 1 if the $x-$coordinate of the projected pixel of the first point is greater than that of the second point. This flag is then fed into the keypoint prediction network.

# 5   Additional Keypoint Characteristics

In addition to the main objectives introduced above, there are common, desirable characteristics of keypoints that can benefit many possible downstream tasks, in particular:

- No two keypoints should share the same 3D location.
- Keypoints should lie within the object's silhouette.

**Separation loss** penalizes two keypoints if they are closer than a hyperparameter $\delta$ in 3D:

$$L_{\text{sep}} \; = \; \frac{1}{N^2} \sum_{i=1}^{N} \sum_{j \neq i}^{N} \max \left( 0, \delta^2 - \|X_i - X_j\|^2 \right) \tag{8}$$

Unlike the consistency loss, this loss is computed in 3D to allow multiple keypoints to occupy the same pixel location as long as they have different depths. We prefer a robust, bounded support loss over an unbounded one (*e.g.,* exponential discounting) because it does not exhibit a bias towards certain structures, such as a honeycomb, or towards placing points infinitely far apart. Instead, it encourages the points to be sufficiently far from one another.

Ideally, a well-distributed set of keypoints will automatically emerge without constraining the distance of keypoints. However, in the absence of keypoint location supervision, our objective with latent keypoints can converge to a local minimum with two keypoints collapsing to one. The main goal of this separation loss is to prevent such degenerate cases, and not to directly promote separation.

**Silhouette consistency** encourages the keypoints to lie within the silhouette of the object of interest. As described above, our network predicts $(u_i, v_i)$ coordinates of the $i^{\text{th}}$ keypoint via a spatial distribution, denoted $g_i(u, v)$, over possible keypoint positions. One way to ensure silhouette consistency, is by *only* allowing a non-zero probability inside the silhouette of the object, as well as encouraging the spatial distribution to be concentrated, *i.e.,* uni-modal with a low variance.

During training, we have access to the binary segmentation mask of the object $b(u, v) \in \{0, 1\}$ in each image, where 1 means foreground object. The silhouette consistency loss is defined as

$$L_{\text{obj}} \; = \; \frac{1}{N} \sum_{i=1}^{N} -\log \sum_{u,v} b(u, v) g_i(u, v) \tag{9}$$

Note that this binary mask is only used to compute the loss and not used at inference time. This objective incurs a zero cost if all of the probability mass lies within the silhouette. We also include a term to minimize the variance of each of the distribution maps:

$$L_{\text{var}} = \frac{1}{N} \sum_{i=1}^{N} \sum_{u,v} g_i(u, v) \left\| [u, v]^\top - [u_i, v_i]^\top \right\|^2 \tag{10}$$

This term encourages the distributions to be peaky, which has the added benefit of helping keep their means within the silhouette in the case of non-convex object boundaries.

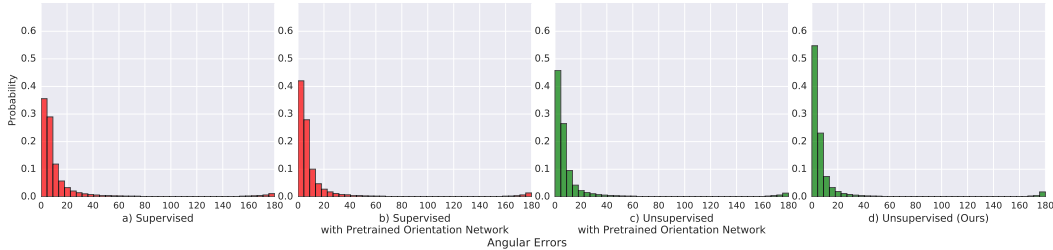

Figure 2: Histogram plots of angular distance errors, average across car, plane, and chair categories, between the ground-truth relative rotations and the least-squares estimates computed from two sets of keypoints predicted from test pairs. a) is a supervised method trained with a single $L_2$ loss between the pixel location prediction to the human labels. b) is the same as a) except the network is given an additional orientation flag predicted from a pretrained orientation network. c) is our network that uses the same pretrained orientation network as b), and d) is our unsupervised method trained jointly (the orientation and keypoint networks).

## 6 Experiments

**Training data.** Our training data is generated from ShapeNet [6], a large-scale database of approximately 51K 3D models across 270 categories. We create separate training datasets for various object categories, including car, chair, and plane. For each model in each category, we normalize the object so that the longest dimension lies in $[-1, 1]$, and render 200 images of size $128 \times 128$ under different viewpoints to form 100 training pairs. The camera viewpoints are randomly sampled around the object from a fixed distance, all above the ground with zero roll angle. We then add small random shifts to the camera positions.

**Implementation details.** We implemented our network in TensorFlow [1], and trained with the Adam optimizer with a learning rate of $10^{-3}$, $\beta_1 = 0.9$, $\beta_2 = 0.999$, and a total batch size of 256. We use the following weights for the losses: $(\alpha_{\text{con}}, \alpha_{\text{pose}}, \alpha_{\text{sep}}, \alpha_{\text{obj}}) = (1, 0.2, 1.0, 1.0)$. We train the network for $200K$ steps using synchronous training with 32 replicas.

### 6.1 Comparison with a supervised approach

To evaluate against a supervised approach, we collected human landmark labels for three object categories (cars, chairs, and planes) from ShapeNet using Amazon Mechanical Turk. For each object, we ask three different users to click on points corresponding to reference points shown as an example to the user. These reference points are based on the Pascal3D+ dataset (12 points for cars, 10 for chairs, 8 for planes). We render the object from multiple views so that each specified point is facing outward from the screen. We then compute the average pixel location over user annotations for each keypoint, and triangulate corresponding points across views to obtain 3D keypoint coordinates.

For each category, we train a network with the same architecture as in Section 4 using the supervised labels to output keypoint locations in normalized coordinates $[-1, 1]$, as well as depths, using an $L_2$ loss to the human labels. We then compute the angular distance error on 10% of the models for each category held out as a test set. (This test set corresponds to 720 models of cars, 200 chairs, and 400 planes. Each individual model produces 100 test image pairs.) In Figure 2, we plot the histograms of angular errors of our method vs. the supervised technique trained to predict the same number of keypoints, and show error statistics in Table 1. For a fair comparison against the supervised technique, we provide an additional orientation flag to the supervised network. This is done by training another version of the supervised network that receives the orientation flag predicted from a pre-trained orientation network. Additionally, we tested a more comparable version of our unsupervised network where we use and fix the same pre-trained orientation network during training. The mean and median accuracy of the predicted orientation flags on the test sets are as follows: cars: (96.0%, 99.0%), planes: (95.5%, 99.0%), chairs: (97.1%, 99.0%).

Our unsupervised technique produces lower mean and median rotation errors than both versions of the supervised technique. Note that our technique sometimes incorrectly predicts keypoints that are $180°$ from the correct orientation due to incorrect orientation prediction.

| Method | Cars Mean | Cars Median | Cars 3D-SE | Planes Mean | Planes Median | Planes 3D-SE | Chairs Mean | Chairs Median | Chairs 3D-SE |
|---|---|---|---|---|---|---|---|---|---|
| a) Supervised | 16.268 | 5.583 | 0.240 | 18.350 | 7.168 | 0.233 | 21.882 | 8.771 | 0.269 |
| b) Supervised with pretrained O-Net | 13.961 | 4.475 | 0.197 | 17.800 | 6.802 | 0.230 | 20.502 | 8.261 | 0.248 |
| c) Ours with pretrained O-Net | 13.500 | 4.418 | 0.165 | 18.561 | 6.407 | 0.223 | 14.238 | 5.607 | 0.203 |
| d) **Ours** | 11.310 | 3.372 | 0.171 | 17.330 | 5.721 | 0.230 | 14.572 | 5.420 | 0.196 |

Table 1: Mean and median angular distance errors between the ground-truth rotation and the Procrustes estimate computed from two sets of predicted keypoints on test pairs. O-Net is the network that predicts a binary orientation. 3D-SE is the standard errors described in Section 6.1.

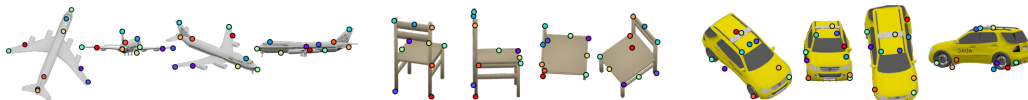

Figure 3: Keypoint results on single objects from different views. Note that these keypoints are predicted consistently across views even when they are completely occluded. (*e.g.,* the red point that tracks the back right leg of the chair.) Please see *keypointnet.github.io* for visualizations.

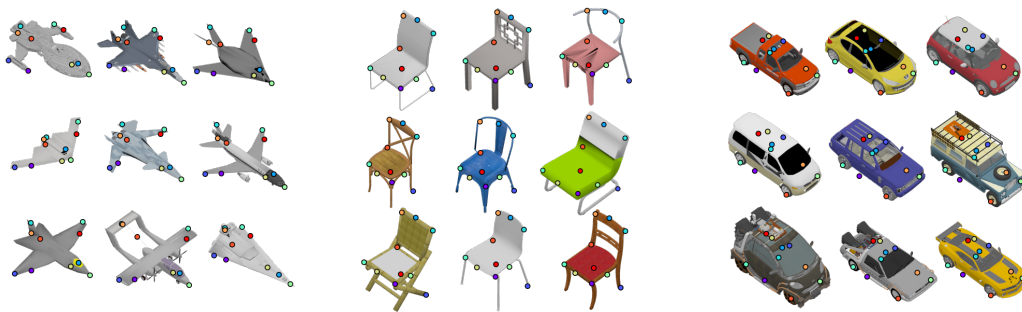

Figure 4: Results on ShapeNet [6] test sets for cars, planes, and chairs. Our network is able to generalize across unseen appearances and shape variations, and consistently predict occluded parts such as wheels and chair legs.

**Keypoint location consistency.** To evaluate the consistency of predicted keypoints across views, we transform the keypoints predicted for the same object under different views to object space using the known camera matrices used for rendering. Then we compute the standard error of 3D locations for all keypoints across all test cars (3D-SE in Table 1). To disregard outliers when the network incorrectly infers the orientation, we compute this metric only for keypoints whose error in rotation estimate is less than $90°$ (left halves of the histograms in Figure 2), for both the supervised method and our unsupervised approach.

## 6.2 Generalization across views and instances

In this section, we show qualitative results of our keypoint predictions on test cars, chairs, and planes using a default number of 10 keypoints for all categories. (We show results with varying numbers of keypoints in the Appendix.) In Figure 3, we show keypoint prediction results on single objects from different views. Some of these views are quite challenging such as the top-down view of the chair. However, our network is able to infer the orientation and predict occluded parts such as the chair legs. In Figure 4, we run our network on many instances of test objects. Note that during training, the network only sees a pair of images of the same model, but it is able to utilize the same keypoints for semantically similar parts across all instances from the same class. For example, the blue keypoints always track the cockpit of the planes. In contrast to prior work [49, 17, 63] that learns latent representations by training with restricted classes of transformations, such as affine or 2D optical flow, and demonstrates results on images with small pose variations, we learn through physical 3D transformation and are able to produce a consistent set of 3D keypoints from any angle.

Our method can also be used to establish correspondence between two views under out-of-plane or even 180° rotations when there is no visual overlap.

**Failure cases.** When our orientation network fails to predict the correct orientation, the output keypoints will be flipped as shown in Figure 5. This happens for cars whose front and back look very similar, or for unusual wing shapes that make inference of the dominant direction difficult.

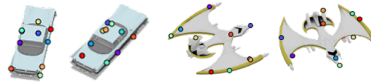

Figure 5: Failure cases.

## 7 Discussion & Future work

We explore the possibility of optimizing a representation based on a *sparse* set of keypoints or landmarks, without access to keypoint annotations, but rather based on an end-to-end geometric reasoning framework. We show that, indeed, one can discover consistent keypoints across multiple views and object instances by adopting two novel objective functions: a relative pose estimation loss and a multi-view consistency objective. Our translation equivariant architecture is able to generalize to unseen object instances of ShapeNet categories [6]. Importantly, our discovered keypoints outperform those from a direct supervised learning baseline on the problem of rigid 3D pose estimation.

We present preliminary results on the transfer of the learned keypoint detectors to real world images by training on ShapeNet images with random backgrounds (see supplemental material). Further improvements may be achieved by leveraging recent work in domain adaptation [24, 54, 50, 5, 58]. Alternatively, one can train KeypointNet directly on real images provided relative pose labels. Such labels may be estimated automatically using Structure-from-Motion [32]. Another interesting direction would be to jointly solve for the relative transformation or rely on a coarse pose initialization, inspired by [55], to extend this framework to objects that lack 3D models or pose annotations.

Our framework could also be extended to handle an arbitrary number of keypoints. For example, one could predict a confidence value for each keypoint, then threshold to identify distinct ones, while using a loss that operates on unordered sets of keypoints. Visual descriptors could also be incorporated under our framework, either through a post-processing task or via joint end-to-end optimization of both the detector and the descriptor.

## 8 Acknowledgement

We would like to thank Chi Zeng who helped setup the Mechanical Turk tasks for our evaluations.

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
