[Supplementary Material]

# Appendix: Discovery of Latent 3D Keypoints via End-to-end Geometric Reasoning

∗ See our supplementary visualizations at *keypointnet.github.io*.

## 1 Histograms for individual categories

We show histograms similar to Figure 2 in the paper for individual object categories.

Figure 1: Histogram plots of angular distance errors between the ground-truth relative rotations and the least-squares estimates computed from two sets of keypoints predicted from test pairs. a) is a supervised method trained with a single $L_2$ loss between the pixel location prediction to the human labels. b) is the same as a) except the network is given an additional orientation flag predicted from a pretrained orientation network. c) is our network that uses the same pretrained orientation network as b), and d) is our unsupervised method trained jointly (the orientation and keypoint networks).

## 2 Ablation study

We present an ablation study for the primary losses as well as how their weights affect the results (Figure 2).

**Removing multi-view consistency loss.** This causes some of the keypoints to move around when the viewing angle changes, and not track onto any particular part of the object. The pose estimation loss alone may only provide a strong gradient for a number of keypoints as long as they give a good rotation estimate, but it does not explicitly force every point to be consistent.

**Pose estimation loss & Noise.** Removing pose estimation loss completely leads the network to place keypoints near the center of an object, which is the area with the least rotation motion, and thus

Figure 2: An ablation study for the losses. a) Our baseline model. b) and c) use twice the noise (0.2) and no noise respectively in the pose estimation loss. d) removes the pose estimation loss. e) removes the silhouette loss.

least pixel displacement under different views. Increasing the noise that is added to the keypoints for rotation estimation encourages the keypoints to be spread apart from the center.

**Removing silhouette consistency.** This causes the keypoints to lie outside the object. Interestingly, the keypoints still satisfy multi-view consistency, and lie on a virtual 3D space that rotates with the object.

# 3   Results on deformed object

To evaluate the robustness of these keypoints under shape variations such as the length of the car, and whether the network uses local features to detect local parts as opposed to placing keypoints on a regular rigid structure, we run our network on a non-rigidly deformed car in Figure 3. Here we show that the network is able to predict where the wheels are and the overall deformation of the car structure.

Figure 3: Results on a non-rigidly deformed car.

# 4   Results using different numbers of keypoints

We trained our network with varying number of keypoints $\{3, 5, 8, 10, 15, 20\}$. The network starts by discovering the most prominent components such as the head and wings, then gradually tracks more parts as the number increases.

Figure 4: Results using networks trained to predict different numbers of keypoints. (Colors do not correspond across results as they are learned independently.)

# 5   Proof-of-concept results on real-world images

To predict keypoints on real images, we train our network by adding random backgrounds, taken from SUN397 dataset [1], to our rendered training examples. Surprisingly, such a simple modification allows the network to predict keypoints on some cars in ImageNet. We show a few hand-picked results as well as some failure cases in Figure 5. The network especially has difficulties dealing with large perspective distortion and cars that have strong patterns or specular highlights.

Figure 5: Proof-of-concept results on real images.

# References

[1] Jianxiong Xiao, James Hays, Krista A Ehinger, Aude Oliva, and Antonio Torralba. Sun database: Large-scale scene recognition from abbey to zoo. *CVPR*, 2010.