[Reviews · NeurIPS 2018]

Reviewer 1



This paper presents a method to learn a set of 3D keypoints which are category-specific using an end to end training. The interesting aspect of this work is that the keypoints position is a latent representation rather than explicit, i.e. we do not need labelling of the positions for training. Rather, the network is trained by provide pairs of images with ground truth poses. - Trying to compute automatically keypoints without annotating them is a great idea and to the reviewer knowledge the first time to be presented. One question that might rises is what happens with objects where is not trivial to annotate points, a discussion would be interesting. - Another point that would require a more consistent discussion is which provide the most keypoint separation i.e. Separation loss and Silhouette consistency. - I would like to comment why it was not possible to train/compare directly with Pascal3D: http://cvgl.stanford.edu/projects/pascal3d.html - It would be interesting to check objects for which keypoints are difficult to extract (spheroids or alike) in terms of failure cases and keypoint extracted (against humans).

Reviewer 2



Overall this is significant and original work for learning 3D keypoints from images. The most significant contribution is the use of a differentiable Procrustes alignment layer, which directly optimizes the keypoints for the upstream task of relative pose estimation. This is a great direction for research in keypoint learning, which often settles for optimizing intermediate metrics such as repeatability, matching score or semantic meaningfulness. The authors use synthetic 3D models with known relative pose ground truth for training, and evaluate also on synthetic shapes. The paper is well-organized with easy to follow arugments + supporting math. One criticism of the high-level approach of the paper is that the keypoints are class specific and don't generalize to an arbitrary number of objects and points the scene. The keypoints also lack descriptors, so it's unclear how this method would deal with aligning more complicated and noisy scenes where there isn't a pre-defined correspondence mapping between points. The proposed solution for dealing with symmetry by using two separate networks at inference time seems inelegant. Given that the two tasks are highly related, the two networks might be combined into a single multi-task network, which implicitly learns the orientation by predicting the orientation flag. It would also be interesting to report the accuracy of the binary flag predictor network. I love the visualizations in the supplemental material website. I found the ablation study visualizations particularly useful in understanding the importance of the various losses used. Adding some numbers to go along with these visualizations would help convey this result in the main paper. Typo/Question: What is the 3D-SE metric? I don't see its definition in section 6.1 as is mentioned in the caption in Table 1.

Reviewer 3



The paper presents a self-supervised approach for detecting 3D key-points using end-to-end geometric reasoning. Given two views of the same object with a known rigid transformation, the network is indirectly trained for keypoint localization by seeking key-points that best relate both views. An objective function is proposed that incorporates multiple domain-specific geometric cues to train the network. Multi-view consistency between the two sets of points under ground-truth transformation is used to enforce that the estimated key-points from both views can be transformed to each other via the known rigid transformation. To avoid degenerated solutions where all key-points collapse to the same location, two additional loss functions are introduced. The first loss measures the misfit between the ground-truth relative rotation and the estimated relative rotation using estimated key-points, while the second loss discourages proximity of the key-points in 3D. Finally, a silhouette consistency loss is introduced to ensure that the 2D projections of 3D key-joints lie inside the object silhouettes. Experiments are performed on the ShapeNet dataset, where the authors have obtained the annotations for three object categories via Mechanical Turk. Experimental evaluation demonstrates that the proposed self-supervised approach outperform supervised counterpart. Pros: + The paper is well-written and easy to read. + The idea of discovering 3D key-point is very nice, though not completely novel (see below) + I like the idea of relative pose estimation as a supervisory signal. + The qualitative results look quite good. + The proposed approach can be extended to real applications such human pose estimation and hand pose estimation. The used supervisory signals for human pose estimation are also available, for example, in HUMAN3.6M dataset or for hand pose estimation in CMU's Panoptic Studio. Cons: - The idea of self-supervised 3D keypoint localization is not entirely novel and have been addressed in the literature. For example in [a]. - I would have liked to see quantitative ablative studies in the paper, that how much each loss contributes to the final performance. At the moment this is only provided in the supp. material without any quantification. - I would have liked to see some results on realistic data such as for HUMAN3.6M or some other application where same supervisory signals can be used. Summary: I think it is a good paper with sufficient novel contributions to be accepted to NIPS. The proposed method can be extended to many other applications. Some references that should be added: [a] Tung et al., Self-supervised Learning of Motion Capture, NIPS'17 [b] Rohdin et al., Unsupervised Geometry-Aware Representation for 3D Human Pose Estimation, CVPR'18 [c] Molchanov et al., Improving Landmark Localization with Semi-Supervised Learning, CVPR'18 [d] Zhang et al., Unsupervised Discovery of Object Landmarks as Structural Representations, CVPR'18 or other recent CVPR'18 papers that I might have missed. The rebuttal addressed most of my concerns. It still misses experiments on a real application, but I agree with the authors that it is an interesting future direction.